# Automated Prediction of Bacterial Exclusion Areas on SEM Images of Graphene–Polymer Composites

**DOI:** 10.3390/nano13101605

**Published:** 2023-05-10

**Authors:** Shadi Rahimi, Teo Lovmar, Alexandra Aulova, Santosh Pandit, Martin Lovmar, Sven Forsberg, Magnus Svensson, Roland Kádár, Ivan Mijakovic

**Affiliations:** 1Division of Systems and Synthetic Biology, Department of Life Sciences, Chalmers University of Technology, 41296 Gothenburg, Sweden; shadir@chalmers.se (S.R.);; 2Division of Engineering Materials, Chalmers University of Technology, 41296 Gothenburg, Sweden; 3Wellspect Healthcare, Aminogatan 1, 43121 Mölndal, Sweden; 42D Fab AB, Bultgatan 20, 85350 Sundsvall, Sweden; 5The Novo Nordisk Foundation Center for Biosustainability, Technical University of Denmark, 2800 Kongens Lyngby, Denmark

**Keywords:** antibacterial, bacterial exclusion area, graphene flakes, algorithm, vertical

## Abstract

To counter the rising threat of bacterial infections in the post-antibiotic age, intensive efforts are invested in engineering new materials with antibacterial properties. The key bottleneck in this initiative is the speed of evaluation of the antibacterial potential of new materials. To overcome this, we developed an automated pipeline for the prediction of antibacterial potential based on scanning electron microscopy images of engineered surfaces. We developed polymer composites containing graphite-oriented nanoplatelets (GNPs). The key property that the algorithm needs to consider is the density of sharp exposed edges of GNPs that kill bacteria on contact. The surface area of these sharp exposed edges of GNPs, accessible to bacteria, needs to be inferior to the diameter of a typical bacterial cell. To test this assumption, we prepared several composites with variable distribution of exposed edges of GNP. For each of them, the percentage of bacterial exclusion area was predicted by our algorithm and validated experimentally by measuring the loss of viability of the opportunistic pathogen *Staphylococcus epidermidis.* We observed a remarkable linear correlation between predicted bacterial exclusion area and measured loss of viability (R^2^ = 0.95). The algorithm parameters we used are not generally applicable to any antibacterial surface. For each surface, key mechanistic parameters must be defined for successful prediction.

## 1. Introduction

Misuse of antibiotics and the global rise of antibiotic resistance is heralding a post-antibiotic era in which humanity will need to find alternatives to classical antibiotics. One venue that is being extensively explored is the discovery of new materials resulting in surface structures with antibacterial properties. These comprise natural and artificial fabricated surfaces. Natural biocidal surfaces include insect wings with nanopillars (cicada wing [1,2,3]) and animal skins with nanostructures (shark and gecko skins [4,5]). In addition to these, a variety of chemical and mechanical methods were used to replicate these naturally occurring surfaces [6,7,8,9]. Nanostructures on metals such as Cu, Al, Ti, and Au [10,11,12,13,14,15], TiO_2_ [16], silica and Al_2_O_3_ [17], black silicon [18], polymers [19], and nanocomposites such as Ti/Au [20] and Ag/polylactide [21] are examples of these fabricated surfaces. All these nanostructures have various mechanisms of antibacterial action but summarily result in reduced attachment of bacterial biofilms. The key limitation in this process is our ability to measure the antibacterial efficiency of newly developed surfaces rapidly and reliably. Proper microbiology assays which involve assessment of cell viability are needed for this, and these typically require days to weeks. SEM imaging is a valuable tool to assess surfaces modified by nanostructures. However, the information that we can provide by SEM observation is only qualitative information on the density and positioning of the flakes.

To address this bottleneck, we attempted to develop an automated pipeline for the prediction of antibacterial potential of a surface based on scanning electron microscopy (SEM) images. We propose that in order to be efficient, such algorithms cannot be overly generic and must take into account a certain level of mechanistic understanding of the antibacterial properties of the specific surface. To test this assumption, we developed an example involving a polymer composite containing oriented exfoliated graphite nanoflakes, recently reported by our group [7]. The orientation and density of distribution of exfoliated graphite nanoplatelets within our polymer nanocomposite can be controlled in such a way as to reduce bacterial viability by a factor of 99.9999%, which is currently the benchmark in the field [7]. The key antibacterial property of this material is the density of sharp exposed edges of exfoliated graphite flakes that are accessible on the surface and kill bacteria on contact. The mechanism of interaction of graphene-based materials, including exfoliated graphene, with bacterial cells includes mechanical damage, electron transfer, insertion, lipid extraction, pore formation, and wrapping of cells [22]. Graphene material insertion, lipid extraction, and pore formation are classified as mechanical damage to the cell. Lipid peroxidation and electron transfer cause oxidative stress. Masking mode could also be the underlying basis of the wrapping mechanism.

In general, larger size, sharper edge, and aggregation are advantageous to inserting/cutting, lipid extraction, and pore formation modes, resulting in stronger destabilization of membrane [23]. The size of graphene nanosheets positively correlated with the extent of lipid extraction. Larger graphene flakes had more phospholipid extraction power, thereby demonstrating stronger antibacterial activity compared to smaller flakes [24]. Small-sized graphene materials induce oxidative stress because the density of defects increases with reduction of the size of the materials [23]. The wrapping or masking mechanism also requires large lateral size, specifically, of micrometer-sized graphene materials [25], as smaller graphene could easily pierce the phospholipid membrane [26]. Wrapping can basically be ruled out with nanoflakes embedded in a polymer matrix. The orientation of graphene materials could also be engineered to confer superior antibacterial property [27]. Magnetic field and chemical vapor deposition methods are used to control the orientation of graphene materials [28,29]. Our group demonstrated that vertically aligned graphene materials are more lethal for bacteria than randomly orientated ones [7]. This might be due to the synergistic effect of physical puncturing of cell membrane and effective electron transfer, as vertical graphene and cell membrane are in contact with each other efficiently [30].

In a nutshell, for a particular antibacterial surface to work, the surface area of sharp exposed edges of exfoliated graphite flakes accessible to bacteria needs to be inferior to the diameter of a typical bacterial cell (Figure 1).

To develop and test our predictor of the antimicrobial effectiveness of these polymer nanocomposites, we devised an algorithm that analyzes SEM images of the surface, identifies exposed (accessible) edges of exfoliated graphene, predicts bacterial exclusion areas around the edges (based on bacterial diameter), and calculates the remaining area available to bacterial cells (Figure 1). Next, we prepared six composite samples in which the distribution and orientation of exfoliated graphite nanoflakes varied considerably, and we measured the loss of bacterial viability on each surface. SEM images of those same surfaces were randomly obtained and analyzed by our algorithm. The correlation between predicted bacterial exclusion area and the measured loss of viability for each surface turned out to be proportional, with an R^2^ coefficient of 0.95. An overview of the analysis presented in this study is shown in Figure 1. We concluded that prediction of antibacterial potential from SEM images is possible for this type of material. We argue that our approach is generally applicable; however, for other types of surfaces, the approach will be valid only if the antibacterial mechanism is understood well enough to include the key parameters in the prediction.

## 2. Results and Discussion

### 2.1. Algorithm Predicts Bacterial Exclusion Areas from SEM Images of Composite Surface

We fabricated 6 polymer composite samples with varying density of graphene flakes (5, 10, 15, 20%) sliced in a longitudinal (L) and/or transversal (T) direction (Table 1). For all samples, SEM images were randomly obtained and submitted to computational analysis. To assess the antibacterial property of the samples, software was developed to implement an algorithm for predicting and quantifying the bacteria exclusion area, defined as a fixed radius around each detected graphene edge (Figure 2). The algorithm identifies flake edges (Figure 2a), identifies zones in which flake edges are vertically oriented and exposed enough to have antibacterial effects (Figure 2b), defines the exclusion zone as a perimeter around each edge that is not accessible to the bacteria (Figure 2c), and, finally, predicts the bacterial exclusion area as a % of the surface not accessible to bacteria (Figure 2d). It should be noted that the graphene-integrated area is not necessarily the same as the bacterial exclusion area of the graphene-integrated polymer surface. In fact, the bacterial exclusion area is the area where the attachment of bacteria is impossible due to the exposed edges of exfoliated graphene, and not due simply to the presence of graphene within the polymer. The boundary of bacterial exclusion area is determined based on geometrical assumptions that the bacteria are not able to attach in the vicinity of these exposed sharp graphene edges. Thus, the radius of the nonattachment area is determined based on the average size of bacteria, which is 0.9 µm in the case of *Staphylococcus epidermidis*. The basic concept behind the bacterial exclusion area in the algorithm is that the area with exposed sharp graphene edges is a “killing zone” for bacteria since the graphene edges disrupt the membrane of bacterial cells and thereby confer an antibacterial property to the material. The concept might be generalized to other materials if they exhibit the same mechanism of physical damage to bacterial cells as graphene.

The software is available from this link: https://github.com/SysBioChalmers/bacterial-exclusion-prediction (created on 21 February 2023). There are different versions of the software for different operating systems, such as Linux, MacOS, or Windows, that can be installed as explained in the installation section. The input for the algorithm is an image in “tif” format. The software makes use of Tesseract, a robust text recognition algorithm [31], and can automatically read the text showing the magnification written on SEM images. Thus, the software can adapt the results to the scale and the estimated size, provided in µm, of the flakes.

Several parameters can be controlled in the “Bacteria exclusion” section of the web interface. One can adjust the contrast threshold and the minimum area (pixel^2^) for identifying an edge. The radius around the graphene edge (μm) to exclude bacteria can also be adjusted to accommodate bacterial cells of different sizes or shapes [23]. We recommend using images with magnification between 1000×–2500×. The algorithm can be run in batch mode, allowing the user to specify a directory from which all existing images are analyzed together, and the output is stored inside an automatically generated subdirectory, “output”, so as not to clutter the working directory.

Flake sharpness, properly oriented flakes, and predicted areas contributing to bacterial exclusion from the surface could be visualized in the algorithm output files, including “edge_sharpness.png”, “graphene.png”, and “bacteria-exclusion.png” files, respectively (Figure 2a, Figure 2b, and Figure 2d). The “edge_sharpness.png” is derived by calculating the sum of the absolute contrast for each pixel in each of the four directions (as explained in the section covering the bacterial exclusion area in Materials and Methods). This provides an image of the sharpness at each pixel. The “graphene.png” image is derived from the contrast by using a threshold and then identifying connected regions. These regions, or flakes, are then filtered by minimum area to remove smaller noise. Finally, the “bacteria-exclusion.png” image is created by calculating if the distance to the closest flake is below a threshold or not. The algorithm also visualizes the percentage of excluded bacteria area (the percentage of white pixels in Figure 2d) in relation to the terminal.

### 2.2. Predicted Bacterial Exclusion Areas Correlate Well with Experimental Measurements of S. epidermidis Viability

The percentage of bacterial exclusion areas predicted by the algorithm was compared to measured antibacterial effects for all tested surfaces [7]. While the CFU counting method provides an aggregate measure for the entire tested surface (typically a sample area of 0.5 cm^2^), for practical reasons, the algorithm prediction was obtained from a sample of SEM images that are representative, but do not cover the entire sample surface (5 SEM images from random positions, covering a total area of 0.1–0.2 mm^2^). The average % of bacterial exclusion from these SEM images was compared with the measured aggregate % of loss of viability. As shown in Figure 3, Figure 4 and Figure 5, to obtain the proper bacterial exclusion value, the user is required to adjust the “contrast threshold for a valid edge” to cover all the flakes, as indicated in Table 1. It is recommended to choose a lower contrast threshold, in the range of 7–25, for images with higher magnification (2500×–5000×) (Table 1). However, a higher contrast threshold, of 20–40, would be required for images obtained at lower magnification (500×–1500×) (Table 1).

As we previously investigated [7], the ratio of graphite nanoplatelets to total surface was significantly different among longitudinally (L) and transversally (T) sliced samples with ≥15% integrated graphite nanoplatelets. The ratio of graphite nanoplatelets to total surface was significantly higher in longitudinally (L) sliced samples compared to transversally (T) sliced samples with the same density of graphite nanoplatelets used for fabrication. Subsequently, there was an clear difference in loss of *Staphylococcus epidermidis* viability among these samples, as longitudinally (L) sliced polymers with ≥15% graphite nanoplatelets showed higher loss of viability compared to the transversally (T) sliced samples. Thus, the increased ratio of graphite nanoplatelets to total surface in longitudinally (L) sliced polymers could result in increased loss of viability. Interestingly, these results were clearly reflected by our prediction and, as it was shown (Table 1), the average predicted bacterial exclusion of 15%-L images was 89.86% ± 2.91%, which was significantly higher than that of 15%-T images (73.85% ± 6.84%).

As indicated in Table 1, the average % of bacterial exclusion area predicted by the algorithm matches the trend of the measured loss of viability of *S. epidermidis* cells in all six samples. In addition to the six samples, we have included the first two samples in Table 1. The surfaces of the latter have no proper orientation and show low loss of viability. High heterogeneity on the surface of these samples results in high standard deviation in measured loss of viability; thus, we excluded them from the correlation plot. Based on the results from Table 1, a linear correlation plot was drawn for the predicted bacterial exclusion versus measured loss of viability of *S. epidermidis* (Figure 6). The predicted bacterial exclusion and the measured loss of viability of *S. epidermidis* correlate extremely well, with an R-squared value (R^2^) of 0.95. Our conclusion was that the algorithm can be used to provide reliable estimates of antibacterial protection offered by this specific type of composite surface.

## 3. Conclusions

In this study, we propose that the key parameter for the antibacterial activity of this type of material is the density of distribution of sharp exposed edges of exfoliated graphite flakes that are accessible on the surface and can kill bacteria on contact. Based on this assumption and using SEM images of the surface, we designed an in silico predictor of the antibacterial potential of various exfoliated graphite nanocomposites. The predictor demonstrated a very strong correlation between the actual antibacterial potential of different surfaces (measured experimentally) and the density and orientation of exposed nanoflakes. We propose that the presented approach has the potential to drastically speed up the investigation of new materials with integrations with the potential to prevent bacterial attachment. Thus, it is possible to devise an algorithm that can automatically detect key surface properties from SEM images and, based on this, accurately predict the potential of the surface to offer antibacterial protection. Such in silico predictions based on SEM images can be used to screen a large number of materials before selecting the most promising ones for antibacterial testing (which typically lasts from days to weeks). While we would like to argue that this approach can be of general utility in the field of antibacterial surfaces, we wish to stress that its proper use critically depends on two factors. Firstly, the antibacterial mechanism of the given surface must be sufficiently understood to program the algorithm to recognize the key surface properties. In the presented case, these key features were the orientation and distribution of exposed graphene edges. Secondly, the uniformity of the surface features across the entire sample may play a key role. For materials with less uniform features, it would be advisable to cover larger areas with SEM analysis, which may constitute a tradeoff regarding time (acquisition of SEM images) and computational resources. Within these limits, we would argue that the presented approach has the potential to speed up the investigation of new materials with integrations with the potential to prevent bacterial attachment and curb bacterial infections [4].

## 4. Materials and Methods

### 4.1. Polymer Integration with Graphene Nanoplatelets and Scanning Electron Microscopy (SEM) Imaging

The low-density polyethylene (LDPE) graphene nanoplatelet nanocomposites were prepared as described previously [7]. LDPE was utilized as the composite matrix. M25 graphene nanoplatelets from 2D Fab (Sweden) with average particle diameter of 25 microns and thickness of 6–8 nm were used as a filler. LDPE pellets were cryogenically ground into powder form and mixed with well-dispersed and homogenized suspension of graphene nanoplatelets with acetone. This process was followed by drying at 60 °C. Then, the extrusion process of nanocomposites was carried out using a circular die by Brabender 19/25 D single-screw extruder (Duisburg, Germany) by means of a compression screw (diameter D = 19 mm and screw length of 25 × 19, compression ratio 2:1). Samples for antibacterial analysis and SEM imaging were collected. SEM imaging was performed randomly from the entire surface using Supra 60 VP microscope (Carl Zeiss AG, Oberkochen, Germany).

### 4.2. Evaluation of Antibiofilm Potential

The antibacterial activity of graphene-integrated materials was tested against the opportunistic bacterial pathogen *S. epidermidis* as a model for Gram-positive bacterium [7]. The overnight culture of *S. epidermidis* bacteria was diluted in fresh tryptic soy broth (TSB) (Sigma Aldrich, Stockholm, Sweden) to obtain the final inoculum of 2–5 × 10^6^ CFU/mL and seeded in the pre-sterilized integrated and nonintegrated surfaces. Samples with bacterial inoculum were incubated at 37 °C for 6 h without agitation for the formation of biofilms. After 24 h of growth, samples were collected in 5 mL of 0.89% of sodium chloride solution for viability test. The biofilms were detached from the surface and homogenized by sonication probe at 10% for 30 s. The homogenized biofilm suspensions were serially diluted into 0.89% of sodium chloride solution and plated onto the LB agar plates. Agar plates were incubated at 37 °C for 24 h and the number of colonies was counted. The number of colonies grown on integrated surfaces divided by the number of colonies grown on control nonintegrated surfaces multiplied by 100 were expressed as the percentage of viability; 100 minus the percentage of viability equals percentage of loss of viability.

### 4.3. Development of SEM Image Analysis Algorithm

A software implementing an algorithm for analyzing SEM images from surfaces with graphene orientation was developed. The algorithm was written in the Rust programming language (rust-lang.org) (accessed on 20 June 2022). The algorithm is packed with a graphical user interface for simple parameter testing and a batch analysis feature for efficient screening through a web interface which can be reached through http://127.0.0.1:8080 once the algorithm is up and running. The software is available from this link: https://github.com/SysBioChalmers/bacterial-exclusion-prediction (created on 21 February 2023).

### 4.4. Bacterial Exclusion Area Determination

The source pictures for the analysis were obtained by means of SEM at the fixed settings. Due to different surface topology and conductivity (due to different conductive filler dispersion), the pictures had different values of contrast and brightness. All source pictures were provided in 8-bit pixel form (values ranging from 0 to 255).

The bacterial exclusion area is the area where the attachment of bacterial film is impossible. It is determined, based on geometrical assumptions, that the bacteria will not be able to attach in the vicinity of the sharp graphene edge. The radius of nonattachment is determined by the average size of bacteria; for example, *S. epidermidis* size was determined as 0.9 µm.

The algorithm operates as following:

1. Graphene edge detection. This procedure searches for the “sharp” contrast area in the image. Pixel intensities, *I*, of surrounding pixels were obtained and the absolute intensity difference of pixels positioned opposite from each other with respect to the analyzed pixel was averaged as shown in Figure 7. Value ranging from 0 to 255 was obtained for each pixel, which represents the “absolute” gradient of the intensity of the image and is denoted as *C* further on:
(1)$$C=\frac{1}{4}(\left|I_{N}-I_{S}\right|+\left|I_{W}-I_{E}\right|+\left|I_{NW}-I_{SE}\right|+\left|I_{NE}-I_{SW}\right|)$$


2. Thresholding is carried out by filtering pixels with a preset value of intensity $C_{0}$. This value can be set by user. Binary image is the output of this operation.

3. Denoising is performed according to the following subroutine:
a.The binary image enables us to define groups of connected white pixels isolated from other pixels by black pixels as individual groups. Each group represents a detected graphene flake.b.Calculation of the area of each group by counting the pixels in the group.c.Remove group if the area is below a threshold set by the user.


4. Calculation of the Euclidean distance for every pixel of the image to the closest white pixel. Next, the threshold determined by the size of the bacteria is applied to the calculated value. This operation results in a binary image.

## Figures and Tables

**Figure 1 nanomaterials-13-01605-f001:**
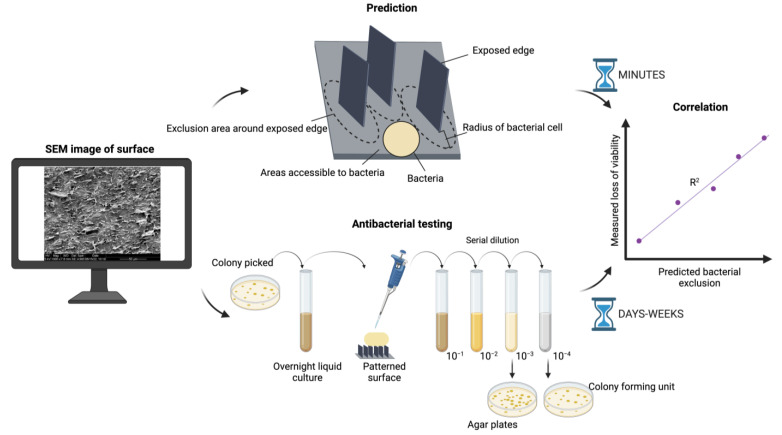
Overview of our comparison of prediction with experimental results. SEM image analysis of the materials is the input. Antibacterial testing, based on counting of colony-forming units (CFUs), can require days to weeks. Prediction based on SEM image analysis and assessment of bacterial exclusion area is carried out in minutes. Predicted and measured antibacterial effect for each tested surface is compared and correlated. Figure was created with BioRender.com.

**Figure 2 nanomaterials-13-01605-f002:**
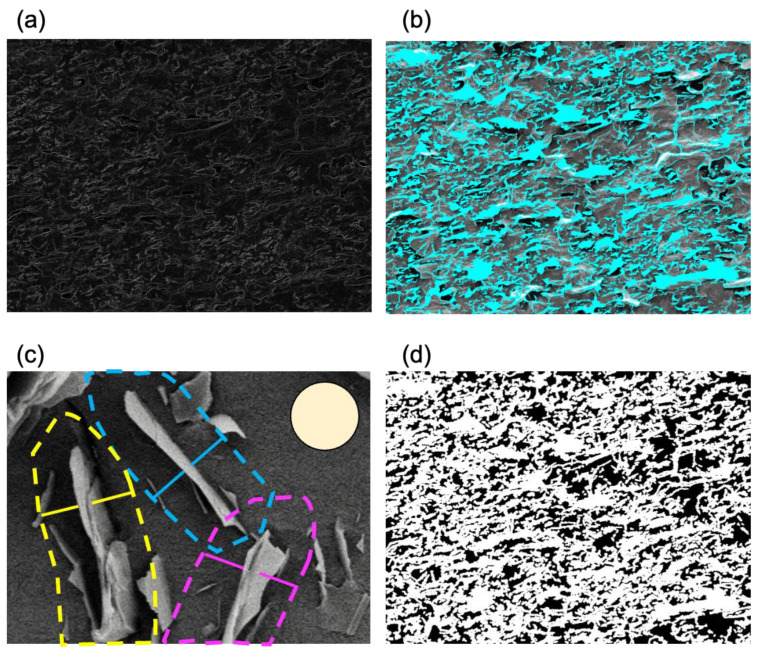
Automated prediction of bacterial exclusion areas. (**a**) Detection of graphene flake edges, (**b**) definition of areas with correct graphene orientation (marked in blue), (**c**) the exclusion zone around each edge (dotted line with exclusion radius = 0.9 μm) that is not accessible to the bacteria (tan disc shows a schematic *S. epidermidis* cell with 1.5 μm diameter), and (**d**) predicted bacterial exclusion areas (marked in white) on the analyzed surface.

**Figure 3 nanomaterials-13-01605-f003:**
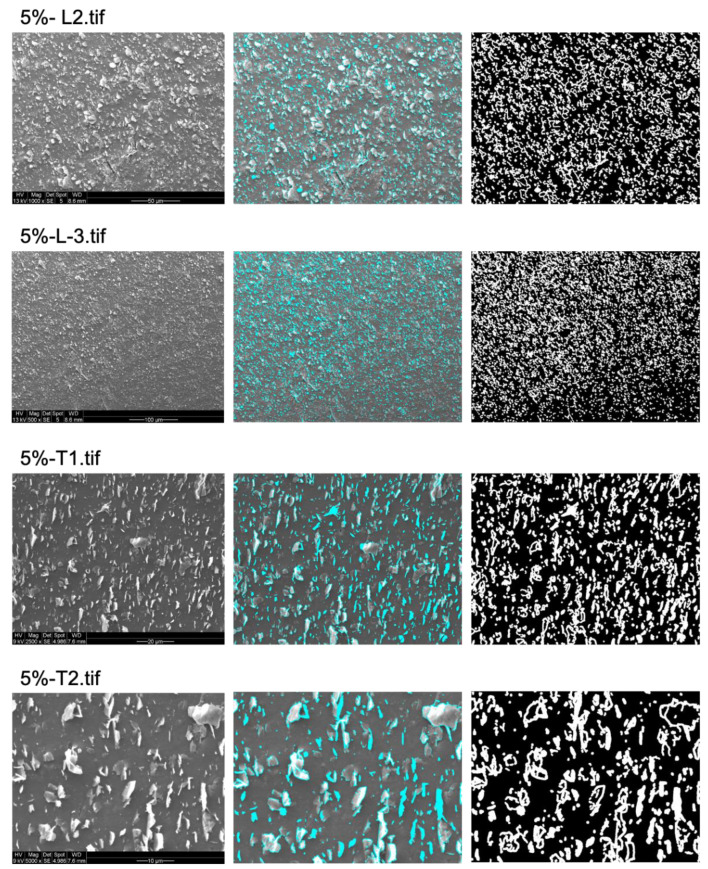
Prediction of areas contributing to bacterial exclusion from 2 different images at different magnifications from 5%-L and 5%-T samples. Proper graphene-oriented areas on original image (**left**) marked in cyan (**middle**) and predicted bacterial exclusion areas marked in white (**right**) on the surface of samples. The original images were adapted with permission from Ref. [7]. 2023, Shadi Rahimi.

**Figure 4 nanomaterials-13-01605-f004:**
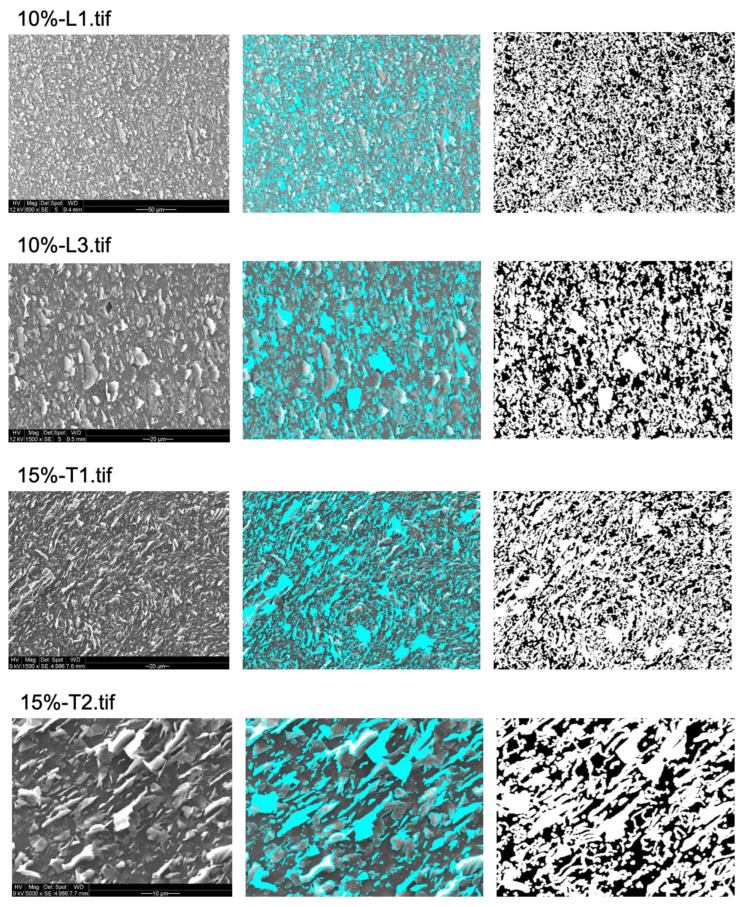
Prediction of areas contributing to bacterial exclusion from 2 different samples of 10%-L and 15%-T. Two images at random locations and at different magnifications were obtained from each sample. Original image is shown on left, proper graphene-oriented areas on the surface of samples are marked in blue in middle image, and predicted bacterial exclusion areas are marked in white on the right side. The original images were adapted with permission from Ref. [7]. 2023, Shadi Rahimi.

**Figure 5 nanomaterials-13-01605-f005:**
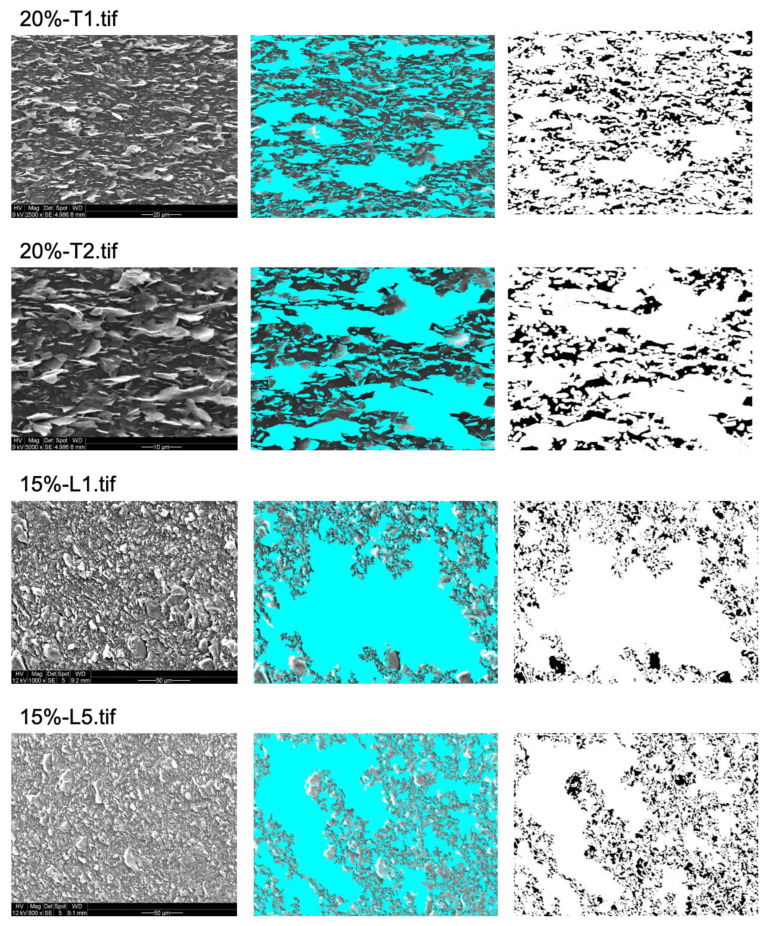
Prediction of areas contributing to bacterial exclusion from 2 different samples of 20%-T and 15%-L. Two images at random locations and at different magnifications were obtained from each sample. Original image is shown on left, proper graphene-oriented areas on the surface of samples are marked in blue in middle image, and predicted bacterial exclusion areas are marked in white on the right side. The original images were adapted with permission from Ref. [7]. 2023, Shadi Rahimi.

**Figure 6 nanomaterials-13-01605-f006:**
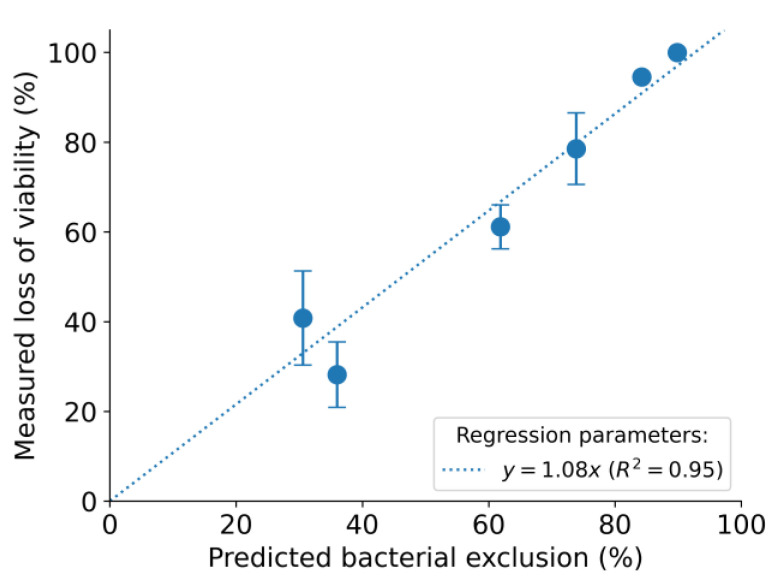
The linear correlation plot for the predicted bacterial exclusion versus measured loss of viability of *S. epidermidis.*

**Figure 7 nanomaterials-13-01605-f007:**
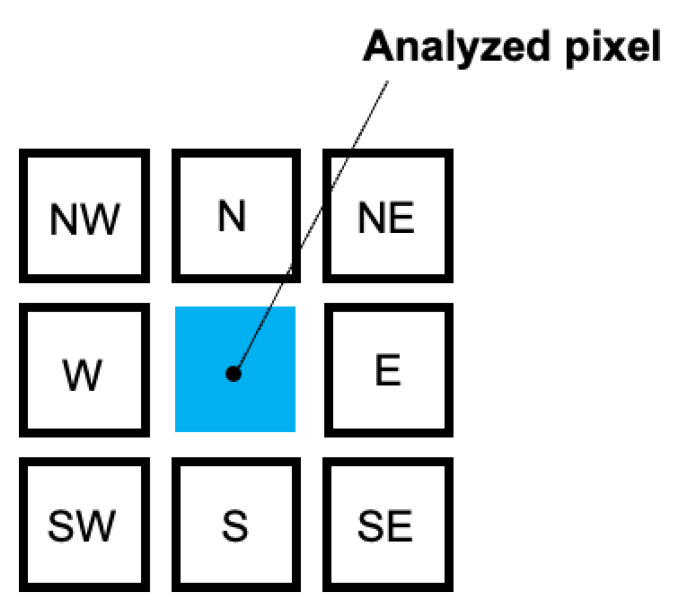
Schematic representation of intensity calculation for edge detection.

**Table 1 nanomaterials-13-01605-t001:** Comparing loss of viability of *S. epidermidis* from entire surface with varying density of graphene flakes (5, 10, 15, 20%) sliced in longitudinal (L) and/or transversal (T) direction. The average of predicted bacterial exclusions was calculated from several images characterized by graphene flakes randomly obtained at different magnifications. The generated outputs were measured using these parameters: minimum_edge_area = 5 and exclusion_radius = 0.9. SEM images were obtained from random positions on 8 different samples. The first four images from two samples have no proper orientation. The loss of viability of *S. epidermidis* measured from whole surface of sample was obtained from our previous study [7]. The data represent the mean ± SD of independent replicates with repetitions.

Image	Contrast Threshold for a Valid Edge	Average of Predicted Bacterial Exclusion from Images from Different Positions in One Sample (%)	Magnification	Measured Loss of Viability (%)
M25-6.tif	50	9.73 ± 1.45	1000×	−16 ± 42.4
M25-2.tif	50		1000×	
23.tif	80	20.61 ± 5.62	1000×	13.518 ± 19.49
11.tif	90		1400×	
5%-L2.tif	40	35.97 ± 3.7	1000×	28.18 ± 7.3
5%-L-3.tif	40		500×	
5%-L-5.tif	40		650×	
5%-L1.tif	40		1000×	
5%-L4.tif	20		1000×	
5%-T1.tif	25	30.58 ± 0.58	2500×	40.80 ± 10.48
5%-T2.tif	15		5000×	
5%-T4.tif	20		5000×	
10%-L1.tif	30	61.85 ± 4.46	800×	61.12 ± 4.9
10%-L2.tif	28		800×	
10%-L3.tif	20		1500×	
10%-L4.tif	20		1500×	
15%-T1.tif	25	73.85 ± 6.84	1500×	78.54 ± 7.98
15%-T2.tif	10		5000×	
15%-T3.tif	25		1000×	
15%-T4.tif	7		5000×	
15%-T6.tif	12		2500×	
20%-T1.tif	15	84.23 ± 5.33	2500×	94.51 ± 0.66
20%-T2.tif	8		5000×	
20%-T3.tif	23		1000×	
20%-T4.tif	10		2500×	
20%-T5.tif	8		5000×	
15%-L1.tif	30	89.86 ± 2.91	1000×	99.99 ± 0
15%-L2.tif	30		1000×	
15%-L3.tif	22		1200×	
15%-L4.tif	20		1200×	
15%-L5.tif	25		800×	

## Data Availability

The software is available from this link: https://github.com/SysBioChalmers/bacterial-exclusion-prediction (created on 21 February 2023).

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
