# Peer review of "Automated Prediction of Bacterial Exclusion Areas on SEM Images of Graphene–Polymer Composites"

_nanomaterials, 2023, doi:10.3390/nano13101605_

Round 1

Reviewer 1 Report

This manuscript is a very interesting report on the relationship between SEM-analyzed graphene-polymers and bacterial exclusion. Therefore, it is a report worthy of publication in Nanomaterials. However, I would like two major revisions.

1.    Fig.S1 and Fig.S2: Unclear

2.    Do you have any references for graphene etc. with different sizes? I would like to know more about the benefits.

Author Response

Question: 1. Fig.S1 and Fig.S2: Unclear

Response: According to reviewer’s comment, Fig.S1 and Fig.S2 were changed to Fig. 4 and Fig. 5 and the content was more clearly explained in figure captions.

Question: 2. Do you have any references for graphene etc. with different sizes? I would like to know more about the benefits.

Response: According to reviewer’s comment, we have added the explanation and references for the antibacterial effect of graphene with different sizes in the introduction part.

Reviewer 2 Report

Automated prediction of bacterial exclusion areas on SEM im ages of graphene-polymer composite

Excellent work, it fully meets all the magazine's criteria,

- Correctness of assumptions

- Appropriateness of the experiments but also analysis of the results

- Detailed description of the experimental procedures

You should pay attention to the following points

- You should pay attention to English terminology

- It would be better not to mention that he discovered a new strategy

- The authors should state the novelty of their work more precisely in the conclusion part

- It should be mentioned that the polymer has been modified with graphene

Author Response

Question: - You should pay attention to English terminology

Response: We have carefully checked the English terminology. We have replaced the graphene coated material with the graphene integrated material within whole manuscript.

Question: - It would be better not to mention that he discovered a new strategy

Response: According to reviewer’s comment, we have removed the mention of discovery of a new strategy. Instead, we refer to our work as an investigation.

Question: - The authors should state the novelty of their work more precisely in the conclusion part

Response: According to reviewer’s comment, we have clearly stated the novelty of our work in the conclusion part.

Question: - It should be mentioned that the polymer has been modified with graphene

Response: We have replaced the graphene coated material with the graphene integrated material within whole manuscript, to further clarify this point.

Reviewer 3 Report

In this manuscript, authors used low-density polyethylene (LDPE)-graphene nanoplatelet nanocomposites with different graphene nanoplatelet contents or coatings to investigate the potential of surface property for offering antibacterial protection through SEM images. This study is certain significant and interesting, some issues need to be addressed before it is considered for publication in Nanomaterials.

1. Authors need to cite more references that are relative to the present study in Introduction section.

2. How to determine the boundary of bacterial exclusion area according to different materials. In this work, bacterial exclusion areas seem to equal to coating area of graphene on polymer.

3. Although authors have mentioned six polymer composite samples with varying density of graphene flakes (5, 10, 15, 20%) sliced in longitudinal (L) and/or transversal (T) direction, what is the significance in longitudinal (L) and/or transversal (T) direction during measurement such as loss of viability? How do authors obtain these data in Table 1? It is difficult to understand.

4. Graphene area seems to become bacterial exclusion area, why? If graphene coating is replaced by other material, what’s happened? Some basic principle should be introduced.

5. How to explain the difference of data in L and T directions in Table 1?

I would like recommend authors to carefully revise this manuscript.

Author Response

Question: 1. Authors need to cite more references that are relative to the present study in Introduction section.

Response: According to reviewer’s comment, we have added more references in the introduction part.

Question: 2. How to determine the boundary of bacterial exclusion area according to different materials. In this work, bacterial exclusion areas seem to equal to coating area of graphene on polymer.

Response: According to reviewer’s comment, we have added this text about the determination of the boundary of bacterial exclusion to the manuscript, “It should be noted that the graphene integrated area is not necessarily same as bacterial exclusion area at the graphene integrated polymer surface. In fact, the bacterial exclusion area is the area where the attachment of bacteria is impossible, that is due to the exposed edges of exfoliated graphene, and not due to simply the presence of graphene within the polymer. The boundary of bacterial exclusion area is determined based on geometrical assumptions that the bacteria are not able to attach in the vicinity of these exposed graphene sharp edges. Thus, the radius of non-attachment area is determined based one the average size of bacteria, which is 0.9 µm in case of Staphylococcus epidermidis.” Please refer to Fig 2c for a graphical representation of the bacterial exclusion area.

Question: 3. Although authors have mentioned six polymer composite samples with varying density of graphene flakes (5, 10, 15, 20%) sliced in longitudinal (L) and/or transversal (T) direction, what is the significance in longitudinal (L) and/or transversal (T) direction during measurement such as loss of viability? How do authors obtain these data in Table 1? It is difficult to understand.

Response: According to reviewer’s comment, to make it clear, we have added this text to the manuscript,As we previously investigated [7], the ratio of graphite nanoplatelets to total surface was significantly different among longitudinally (L) and transversally (T) sliced samples with ≥15% integrated graphite nanoplatelets. The ratio of graphite nanoplatelets to total surface was significantly higher in longitudinally (L) sliced samples compared to the transversally (T) sliced ones, with the same density of graphite nanoplatelets used for fabrication. Subsequently, there was an obvious difference in loss of Staphylococcus epidermidis viability among these samples, as longitudinally (L) sliced polymers with ≥15% graphite nanoplatelets showed higher loss of viability compared to the transversally (T) sliced ones. Thus, the increased ratio of graphite nanoplatelets to total surface in longitudinally (L) sliced polymers could result in increased loss of viability. Interestingly, these results were clearly reflected by our prediction and as it was shown (Table 1), the average predicted bacterial exclusion of 15%-L images was %89.86±2.91 which was significantly higher than that of 15%-T images (%73.85±6.84).”

Since we took the loss of viability data from our previous study (Pandit et al., 2020), we have added this information to the table 1 caption: “The loss of viability of S. epidermidis measured from whole surface of sample was taken from our previous study [7].”.

Question: 4. Graphene area seems to become bacterial exclusion area, why? If graphene coating is replaced by other material, what’s happened? Some basic principle should be introduced.

Response: As we have replied to question 2, the graphene integrated area is not necessarily same as bacterial exclusion area at the graphene integrated polymer surface. According to reviewer’s comment, we also clarified the basic concept behind the bacterial exclusion area by addition of this text to the manuscript, “The basic concept behind the bacterial exclusion area in the algorithm is that the area with exposed sharp graphene edges is a “killing zone” for bacteria, since the graphene edges disrupt the membrane of bacterial cells and thereby confer antibacterial property to the material. The concept might be generalized to other materials, if they exhibit the same mechanism of physical damage to bacterial cells as graphene.”

Question: 5. How to explain the difference of data in L and T directions in Table 1?

Response: Please refer to response of question 3, that explains that the difference of data in L and T directions in Table 1. It comes from the differences in ratio of graphite nanoplatelets to total surface in longitudinally (L) and transversally (T) sliced samples.

Round 2

Reviewer 1 Report

This journal operates an open review policy, so your review report may be published. By signing your report your name will appear alongside the published report.

Reviewer 3 Report

Based on the quality of revised manuscript, I would like to recommend this work for publication in Nanomaterials in present form.